# Predicting Antimicrobial Peptide Activity: A Machine Learning-Based Quantitative Structure–Activity Relationship Approach

**DOI:** 10.3390/pharmaceutics17080993

**Published:** 2025-07-31

**Authors:** Eliezer I. Bonifacio-Velez de Villa, María E. Montoya-Alfaro, Luisa P. Negrón-Ballarte, Christian Solis-Calero

**Affiliations:** Faculty of Pharmacy and Biochemistry, Universidad Nacional Mayor de San Marcos, Lima 15001, Peru

**Keywords:** antimicrobial peptides, machine learning, QSAR, classification models

## Abstract

**Background**: Peptides are a class of molecules that can be presented as good antimicrobials and with mechanisms that avoid resistance, and the design of peptides with good activity can be complex and laborious. The study of their quantitative structure–activity relationships through machine learning algorithms can shed light on a rational and effective design. **Methods**: Information on the antimicrobial activity of peptides was collected, and their structures were characterized by molecular descriptors generation to design regression and classification models based on machine learning algorithms. The contribution of each descriptor in the generated models was evaluated by determining its relative importance and, finally, the antimicrobial activity of new peptides was estimated. **Results**: A structured database of antimicrobial peptides and their descriptors was obtained, with which 56 machine learning models were generated. Random Forest-based models showed better performance, and of these, regression models showed variable performance (R^2^ = 0.339–0.574), while classification models showed good performance (MCC = 0.662–0.755 and ACC = 0.831–0.877). Those models based on bacterial groups showed better performance than those based on the entire dataset. The properties of the new peptides generated are related to important descriptors that encode physicochemical properties such as lower molecular weight, higher charge, propensity to form alpha-helical structures, lower hydrophobicity, and higher frequency of amino acids such as lysine and serine. **Conclusions**: Machine learning models allowed to establish the structure–activity relationships of antimicrobial peptides. Classification models performed better than regression models. These models allowed us to make predictions and new peptides with high antimicrobial potential were proposed.

## 1. Introduction

The growing global crisis of resistance to conventional antibiotics represents one of the most urgent threats to public health, generating, year after year, significant societal costs linked to clinical care [1,2]. In this scenario, the search and development of new antimicrobial agents with novel mechanisms of action has become a critical priority. Antimicrobial peptides (AMPs), key components of the innate immune system present in virtually all life forms, have emerged as highly promising therapeutic candidates [3,4].

Despite their vast potential, the discovery and optimization of effective and safe AMPs faces significant challenges. The relationship between amino acid sequence, three-dimensional structure, and antimicrobial activity is intricate and nonlinear. The synthesis and experimental evaluation of the immense diversity of possible peptides are slow, expensive, and low-throughput processes [5]. Therefore, computational methodologies become indispensable to accelerate this process, allowing the screening, prediction, and design of candidates with a higher probability of success before experimental validation [6,7,8,9].

Quantitative Structure–Activity Relationships (QSARs) offer a powerful framework for correlating the physicochemical and structural properties (molecular descriptors) of peptides with their biological (in this case, antimicrobial) activity. By building predictive models, QSAR facilitates the efficient exploration of peptide chemical space [10]. The application of machine learning (ML) techniques to QSAR has revolutionized the field, allowing the modeling of the complex nonlinear interactions that characterize the structure–activity relationship in biomolecules such as AMPs. Advanced ML algorithms, such as Support Vector Machines, Random Forests, or even Deep Learning for sequences, they are particularly suited to handling the high dimensionality of peptide descriptors and building robust models to predict antimicrobial activity [9,11,12,13]. Beyond prediction accuracy, these methods enable mechanistic insights into bioactivity, as demonstrated in network pharmacology studies of compound-target interactions [14] and clinical risk stratification for complex diseases [15].

Based on the above, the main objective of this work is to develop robust machine learning models to establish reliable QSAR for a set of antimicrobial peptides [16]. Beyond the construction of high-performance predictive models, a central focus of our research is the identification and analysis of the most important molecular descriptors that determine the antimicrobial activity of these peptides. Employing these machine learning algorithms, we seek to unravel the key physicochemical and structural features that drive antimicrobial potency, providing fundamental insights for the de novo design or rational breeding of peptides with optimized activity and selectivity profiles in the fight against antimicrobial resistance.

## 2. Materials and Methods

### 2.1. Data Collection

Antimicrobial activity information and linear peptide sequences were collected from the curated databases: DBAASP [17] and APD3 [18]. The information from these databases is condensed since each record corresponds to a peptide, regardless of whether this same peptide has been evaluated against multiple microorganisms. Therefore, an antimicrobial peptide database was generated from these data considering that each record corresponds to a peptide against a specific microorganism and the measurement of its antimicrobial activity.

### 2.2. Calculation of Descriptors

To quantitatively represent the physicochemical properties of the analyzed peptides, molecular descriptors were calculated from previously defined amino acid scales. Each peptide was transformed into a single numerical value, representative of its descriptor profile, using a sliding window approach. Briefly, the calculation of the global descriptor for a peptide (p) of length l(p) was performed as follows: a fixed-size analysis window v was applied that moved along the sequence, generating a total of Nv possible windows. For each window (i), the mean descriptor value $\bar{V_{i}}$ was calculated as the average of the $f_{\left(aj\right)}$ values corresponding to the amino acids contained in that window. The function $f_{\left(aj\right)}$ represents the descriptor index on the selected scale for the amino acid at position j. The final value of the molecular descriptor for the peptide (Dp) was obtained by averaging the values $\bar{V_{i}}$ of all the generated windows, according to the following equations:(1)Nv=l(p)−v+1(2)$$D_{p}=\frac{\sum_{i=1}^{Nv}\bar{V_{i}}}{Nv},\;\text{where}\;\bar{V_{i}}=\frac{\sum_{j=i}^{i+v}{\;f}_{\left(a_{j}\right)}}{v}$$

This procedure allows to obtain a robust numerical representation of the analyzed property, by incorporating both the composition and the local distribution of the amino acids along the peptide sequence. The calculation of these descriptors was performed using the Python programming language (version 3.11.13), employing the modlAMP peptide analysis package [19]. A total of 321 descriptors were calculated, which were selected from the AAIndex database (v9.2) and the modlAMP analysis package (v4.0).

### 2.3. Dataset

#### 2.3.1. Preprocessing

Only peptide records reporting antimicrobial activity as minimum inhibitory concentration (MIC) were selected. To ensure consistency, all MIC values were standardized to the same unit (µg/mL). Subsequently, the data were transformed to logMIC values to normalize their distribution, reduce skewness, and improve the performance of statistical and machine learning models, which often assume normally distributed input variables.

#### 2.3.2. Data Preparation for Regression Models

A single peptide can exhibit multiple antimicrobial activity scores (MIC values) when tested against different microorganisms or reported by various authors under different experimental conditions. To obtain a representative and consistent value for each peptide, the average of all reported logMIC values was calculated. This approach assumes that while the antimicrobial activity can vary depending on the target organism, the peptide’s intrinsic antimicrobial potential remains comparable across contexts. Averaging reduces noise from experimental variability and allows the model to focus on sequence-dependent properties rather than microorganism-specific effects. To mitigate the influence of extreme values, outliers were identified and removed using the interquartile range method.

#### 2.3.3. Data Preparation for Classification Models

Each peptide record was labeled against each of the microorganisms against which they were experimentally evaluated based on their MIC following a pre-established convention: (i) Those with MIC values lower than 25 µg/mL were labeled as “active peptide” (logCMI < 1.3979). (ii) On the other hand, those with MIC values equal to or greater than 100 µg/mL were labeled as “inactive peptide” (logCMI > 2) [20]. Likewise, since the same peptide can be active against some microorganisms and inactive against others, each peptide was labeled as antimicrobial (AMP or Class 1) if it is an “active peptide” against 50% or more of the microorganisms against which it was evaluated and as non-antimicrobial (NO-AMP or Class 0) if it was active against less than 50% of the microorganisms against which it was evaluated. Additionally, some peptides, considered non-antimicrobial, were added to balance the datasets [21]. This classification allows us to distinguish between molecules that have high antibacterial activity and those that do not have it or have a lower or limited activity.

#### 2.3.4. Separation of Subsets

Following the data preparation criteria and from the newly generated database, peptide datasets were extracted against all microorganisms, Gram-positive bacteria, Gram-negative bacteria, *Escherichia* genus, *Pseudomonas* genus, *Staphylococcus* genus and *Bacillus genus*. Finally, 14 datasets were obtained, 7 for regression and 7 for classification. All datasets were separated and isolated into training and testing, in a ratio of 8:2.

### 2.4. Generation of Learning Models

#### 2.4.1. Descriptor Selection and Training

Given this potential predictor redundancy, it was decided to randomly retain only one of the predictors, and a correlation analysis was performed to identify highly similar predictors (Pearson correlation R > 0.8). Low-variance descriptors were also eliminated since they do not provide information that differentiates the peptides. To meet the mechanistic objective of the study (interpreting biological activity based on important properties), it was decided to reduce the number of predictors. Briefly, the first stage focused on applying the Recursive Variable Elimination (RFE) technique. This methodology iteratively eliminates the least relevant features. To do this, the RFE algorithm based on Random Forest was used with 5-iteration cross-validation, repeated 5 times. In the second stage, optimization using a genetic algorithm was used to identify the optimal combination of predictors. This process was also based on the Random Forest algorithm and was carried out with a 5-iteration cross-validation. The genetic algorithm was run over 100 to 200 generations, with 20 individuals per generation, an 80% crossover probability, and a 7% mutation probability.

Once the most appropriate descriptors were selected, various algorithms were trained. These algorithms included Multiple Linear Regression (MLR), Partial Least Squares Regression (PLS), Logistic Regression, Random Forest, Support Vector Machine, and Gradient Boosting Machine. Each algorithm was trained using the corresponding peptide datasets. To maximize model performance, hyperparameter optimization was performed for each model through an exhaustive search for optimal configurations. This was performed using 5-iteration internal cross-validation with 5 replicates.

#### 2.4.2. Determining the Domain of Applicability

The scope of applicability was determined using the Euclidean distance-based method known as the Local Outlier Factor (LOF), which is based on the k-nearest neighbors algorithm. Therefore, the scope of applicability of the observations was determined by the population density in a multidimensional space. If the population density of an observation is significantly lower than its nearest neighbors, then it corresponds to an outlier because it lies outside the scope of applicability.

### 2.5. Model Validation/Evaluation

Finally, as part of the external validation, the final performance of the models was evaluated using the test sets. For the regression models, the coefficient of determination (R^2^) was used as the main metric, along with other metrics not shown, such as RMSE (root mean square error) and MAE (mean absolute error). For the classification models, the Matthews correlation coefficient (MCC) was used as the main metric, along with other metrics not shown, such as precision (Pr), accuracy (Ex), sensitivity (Sn), and the area under the ROC curve (AUC–ROC).

### 2.6. Evaluation of Predictors

The importance of the descriptors in the models was determined by the permutation method in each of the generated models and in order to compare the contribution of each descriptor, the importance values were scaled, so that a relative importance (RI) value was obtained on the scale of 0 to 100 in each type of model, with 0 being the least relative importance and 100 the greatest relative importance.

### 2.7. Design and Prediction of New AMPs

Peptides were randomly designed using the modlAMP library in Python, with lengths ranging from 5 to 50 amino acids, excluding cysteine and methionine for synthesizable reasons [19]. Repeated or existing sequences in the database were removed. Classification and regression models were then applied to predict their activity. Peptides with the highest antimicrobial potential, according to the classification model, were modeled in PEPFOLD 3.0 to predict their secondary structure [22]. Likewise, to characterize these peptides externally, their physicochemical properties were calculated using protein scaling from the Expasy server. ProtScale [23].

### 2.8. Statistical Analysis

For all statistical calculations and data analysis, the programming languages python and R were used with the pandas, numpy, scikit-learn, sciPy, seaborn, matplotlib, tidyverse, rstatix and caret libraries. The Kruskal–Wallis test (α < 0.05) was considered for the comparison between the importance means according to the type of descriptor.

## 3. Results

### 3.1. Dataset Description and Descriptor Calculation

A total of 35,329 records of experimental assays evaluating the antimicrobial activity of 4874 peptides against a wide variety of microorganisms and cell types were collected. Of these records, 28,666 (81.3%) corresponded to evaluations carried out against bacteria, of which 18,428 (52.3%) focused on Gram-negative bacteria and 10,238 (29.0%) on Gram-positive bacteria. In addition, 3605 assays (10.2%) were registered against fungi, 1384 (3.9%) against viruses, and 581 (1.6%) against other organisms such as protozoa and bacteria not classified as Gram-positive or -negative (Figure 1). Additionally, 1031 records (2.9%) were identified that reported antineoplastic activity, an aspect that could represent an interesting field of analysis. When analyzing the distribution of records according to microbial genus, it is observed that the majority were concentrated in the following groups: Staphylococcus with 5961 records (17.5%), *Escherichia* with 5687 (16.7%), *Pseudomonas* with 3871 (11.3%), Candida with 2058 (6.0%), *Bacillus* with 1944 (5.7%), *Salmonella* with 1590 (4.7%), *Klebsiella* with 1200 (3.5%), *Enterococcus* with 1071 (3.1%), *Streptococcus* with 978 (2.9%), *Acinetobacter* with 763 (2.2%), *Micrococcus* with 642 (1.9%), *Vibrio* with 638 (1.9%), *Lentivirus* with 619 (1.8%) and *Listeria* with 565 (1.7%). The remaining 6529 records (19.1%) were distributed among other microbial genera (Figure 1). Regarding the methodologies used to evaluate antimicrobial activity, the Minimum Inhibitory Concentration (MIC) stands out widely, with 22,792 records (71.3%), consolidating its position as the most widely used method. This is followed by the Minimum Bactericidal Concentration (MBC) with 2970 records (9.3%), the Median Inhibitory Concentration (MIC) with 1562 (4.9%), and the Lethal Concentration (LC) with 611 (1.9%). Other diverse methods covered 4040 records (12.6%), reflecting the variety of experimental approaches present in the database (Figure 1).

### 3.2. Generation of Learning Models and Application Domain

#### 3.2.1. Variable Selection

After applying preprocessing with the removal of low-variance and highly correlated descriptors, a refined dataset was obtained, consisting of 218 to 221 predictors, representing the most relevant and non-redundant features for the analysis. Predictor selection using RFE and subsequent genetic algorithm significantly reduced the number of these variables for both the regression and classification models (Table 1).

#### 3.2.2. Training and Validation Results

The classification algorithms logistic regression (LR), support vector machine (SVM), Random Forest (RF) and Gradient were trained. Boosting Machine (GBM) using each of the previously described datasets. Among these, the Random Forest and GBM-based models demonstrated the best performance in cross-validation processes (Appendix A).

Similarly, regression models were trained using partial least squares (PLS), SVM, RF, and GBM algorithms on the corresponding datasets. However, the performance of the regression models was generally poor and inconsistent.

During the validation and prediction process of antimicrobial activity in the test set, the predictive capacity of the regression models was evaluated. Using the Random Forest model, antimicrobial activity of 599 peptides was predicted. The coefficient of determination between the predicted and experimental logMIC values was low (R^2^ = 0.459). However, when applying specific models, an improvement in fit was evident. For example, the model trained with data from Gram-negative bacteria achieved an R^2^ of 0.476, while the model for Gram-positive bacteria showed a lower performance (R^2^ = 0.339). Even more specific models, based on the bacterial genus, showed a better fit. The *Escherichia* genus model achieved an R^2^ of 0.547, followed by the model for *Pseudomonas* (R^2^ = 0.415) and *Bacillus* (R^2^ = 0.574), however the model for *Staphylococcus* was lower (R^2^ = 0.360).

Regarding the classification models, predictions were made using the corresponding test sets for each peptide type, employing models trained with Random Forest. Overall, the classification models showed good performance, with Matthew’s correlation coefficients (MCC) between 0.662 and 0.755, as well as adequate accuracy, precision, and sensitivity values (Table 2). The ROC curves evidence robust performance in all models, highlighting that those trained with more specific datasets obtained better performance (AUC > 0.91) compared to the general models, which is consistent with the other metrics evaluated (Appendix A).

#### 3.2.3. Domain of Applicability

Additionally, as an additional step to the predictions made with the test sets, it was determined whether the observations in this set belonged to this analysis domain. The figure shows the AD scores of the overall classification model, for both the training set and the test set, with some values showing outside the domain (ScoreAD < 0) (Figure 2).

### 3.3. Importance of Descriptors

#### 3.3.1. Importance of Each Descriptor

The importance of peptide descriptors was quite heterogeneous, both in regression and classification models, as well as in models generated in each dataset (Appendix A). However, the physicochemical charge descriptors (Charge, ChargeDensity, charge_acid_1_mean and pI) stand out very notably as the most important in almost all models. Also more relevant are the molecular weight (MW) descriptors, the descriptors of the tendency to form alpha helices (CHOP780212, QIAN880129, among others) and those related to hydrophobicity (FAUJ880112, hoopwoods, among others). To explore the relevance that the models give to these descriptors, the distributions of those descriptors in which we found differences in their means were graphed considering antimicrobial and non-antimicrobial peptides in the general classification model (Figure 3).

#### 3.3.2. Importance of the Descriptor Type

The importance of peptide descriptors was grouped by type to reveal the impact of their nature on the generated models. In the classification models, physicochemical property descriptors were found to have the highest average scores across all models (Table 3 and Figure 4. Similarly, in the regression models, physicochemical property descriptors, hydrophobicity, composition, and other properties were found to have the highest average scores depending on the model (Table 4 and Figure 4).

### 3.4. Prediction and Modeling of New Peptides

#### 3.4.1. Generation of New Peptides and Prediction of Their Antimicrobial Activity

Using the modlAMP peptide analysis package in Python, a total of 999,591 new peptide sequences were generated. Peptides were named with the prefix NPEP (Novel Peptide) followed by a number in order. The probability of these peptides being antimicrobial was then predicted using the general classification model as well as specific Random Forest-based models. In addition, the minimum inhibitory concentration of the generated peptides was estimated using the general regression model and specific Random Forest-based regression models. The peptides with the highest probabilities of being antimicrobial according to the general model were selected as having the highest antimicrobial potential (Table 5, Appendix A).

#### 3.4.2. Modeling of Peptides with the Greatest Antimicrobial Potential

The previously described peptides with the greatest antimicrobial potential were subjected to a structure modeling process using the PEPFOLD3.5 server. This process generated 200 structures for each peptide based on its sequence. The models that obtained lower energy scores according to OPEC (Optimized Potential for Efficient structure Prediction) were considered the most representative. Figure 5 shows these peptides, and they were illustrated taking into account their hydrophobic residues, which are marked in blue. Among them, four peptides exhibit notable secondary structures in the form of alpha helices (NPEP160435, NPEP411179, NPEP251829, NPEP608070), five of them present alpha-helix structures with twisted portions (NPEP857629, NPEP535227, NPEP551925, NPEP785404, NPEP494819), while a last peptide shows a beta-sheet conformation (NPEP90259).

#### 3.4.3. Characterization of the New Peptides

A six-scale amino acid profile was calculated for the modeled peptides, and the average was determined, generating a table with these properties. The properties of these peptides were similar, with a few minor exceptions (Table 6). Furthermore, it is observed that all peptides are cationic with a net charge greater than 9 and a lysine frequency greater than or equal to 5.

## 4. Discussion

Antimicrobial peptides act against bacteria, fungi, and viruses through key properties such as cationic charge, hydrophobicity, secondary structure (α-helix, β-sheet), and amphipathicity, which facilitate membrane interaction and microbial lysis. For instance, positively charged peptides bind to anionic phospholipids on bacterial membranes, while hydrophobicity aids in bilayer penetration [24]. To explore the link between structure and activity, experimental and structural data were compiled into a structured database. The APD3 database [18], established in 2004, was a pioneer in collecting natural antimicrobial peptides, followed by others like CAMPR3 [25] and DBAASP [17], reflecting ongoing advances. This study incorporates manually curated information, including new data such as measurement type, quantitative specificity, and taxonomic details of target microorganisms. Additionally, peptides were structurally represented using molecular descriptors—numerical values derived from physicochemical and structural properties—calculated from side-chain characteristics [26] using tools like *modLamp* and AA-INDEX, enabling precise encoding of peptide composition and shape.

Machine learning models allow establishing relationships between data and a specific dependent variable. This study used such models to analyze the relationship between the chemical structure of peptides (based on flanking residues) and their antimicrobial activity. The independent variables were the molecular descriptors, and the dependent variable was antimicrobial activity, the definition of which is somewhat complex due to the variability in evaluation methods (Figure 1). Therefore, the minimum inhibitory concentration (MIC) was selected as the dependent variable, as it is considered the standard for measuring the antimicrobial activity of peptides [17,18,27]. However, there are microbiological experimental factors not considered in this study, such as incubation times and culture media, under the premise that QSAR can still be established despite these variations [28]. Since the same peptide may be active against some microorganisms but not others, it was necessary to establish a criterion to define whether a peptide is antimicrobial or non-antimicrobial, since machine learning classification models require these classes to be well defined. Thus, peptides were classified considering that AMPs have MICs < 25 µg/mL in more than 50% of the microorganisms evaluated, while the NON-AMPs have MIC > 100 µg/mL in 50% of the microorganisms evaluated [20]. In contrast, other studies do not explain the criteria for this classification, differentiating only between antimicrobial and non-antimicrobial peptides in databases [29,30,31,32], which is not clear or it is assumed that a peptide is antimicrobial if it has activity against at least one microorganism, whatever it may be. To characterize each of the peptides, 318 molecular descriptors were calculated, although not all were useful, since some were redundant (highly correlated) or of low variance, which can affect the efficiency and accuracy of the models [33]. Therefore, these variables were discarded using RFE and genetic algorithms, resulting in different subsets of predictor variables for each organism (Table 1). The domain of applicability (DA) of the QSAR model refers to the chemical space defined by the molecular descriptors and allows the reliability of predictions to be assessed [34,35]. In this study, the k-nearest neighbors (KNN) method was used, which considers the variance and covariance of the dataset to ensure that predictions are made on chemically similar molecules [35,36]. Some studies do not consider DA, which limits the implementation of predictive models [20,29,37]. Although there is no consensus on the best methodology to determine it, distance-based techniques are the most commonly used [35]. For example, Tian et al. used Euclidean distances for HIV-1 antiviral peptides [38], and Pinacho-Castellanos et al. employed a consensus of five methods in AMBIT Discovery for antimicrobial peptides [21]. Being outside the applicability domain (AD) does not invalidate a prediction, but it does reduce its reliability, and therefore such predictions should be treated with greater caution.

In this study, 28 regression and 28 classification models were generated based on four algorithms (Appendix A), trained with physicochemical, structural, hydrophobic, and compositional descriptors calculated from the amino acid sequence. Random Forest (RF)-based models were the best performers in both regression and classification. However, the regression models showed limited performance on the test set (R^2^ = 0.33–0.57), with better results (in some cases) when training was restricted to specific bacterial groups (Table 2). Wang et al. [39] multiple regression models for three classes of antimicrobial peptides, achieving R^2^ of 0.326, 0.589 and 0.663, using 89 descriptors. Their peptides were homogeneous in length (9 or 12 residues), unlike this work that considered peptides of any length. On the other hand, Avram et al. [40] used only eight descriptors for 37 mastoparan-derived peptides, obtaining R^2^ between 0.655 and 0.720, possibly due to the high similarity between the analyzed peptides. In contrast, this study is the first to apply regression models to a large and diverse set of peptides, which increases the complexity of the analysis, but allows to address greater structural and biological variability.

The classification models showed good performance (ACC ≥ 0.831 and MCC ≥ 0.662), also improving when using data restricted by bacterial (Table 2). Pinacho-Castellanos et al. [21] developed five RF models with 96,026 descriptors, achieving an ACC of 0.90 with 135 predictors. This study obtained an ACC of 0.831 using only 26 predictors, facilitating the interpretation of the structure–activity relationship. Vishnepolsky et al. [20] used the DBSCAN algorithm and nine redesigned molecular descriptors to differentiate peptides against Gram-negative bacteria, achieving ACC = 0.80 ± 0.02, slightly lower than that of this study. Here, the optimization and selection of a larger set allowed a more precise characterization, while also searching for a relationship with their antimicrobial activity. Dong et al. [41] acid-based (RAAC) descriptors to classify peptides according to their target microorganism, achieving high ACCs for parasites, viruses, and cancer (89.72–91.92), but lower performance for fungi and Gram-positive and -negative bacteria (74.73–77.92). In comparison, the models in the present study showed superior performance, indicating that compositional descriptors may not be sufficient to classify antimicrobial peptides against bacteria due to their higher complexity and variability.

After model generation, the relative importance of each descriptor, both individually and by type, was assessed in the classification and regression models. In the classification models, all descriptors contribute to overall performance; however, those related to physicochemical properties are, on average, most relevant. Similarly, in the regression models, physicochemical descriptors showed the highest relative importance, although certain descriptors related to hydrophobicity, alpha-structure propensity, and composition were also identified as relevant. Among the physicochemical descriptors, easy-to-interpret variables such as molecular weight (MW), isoelectric point (pI), and peptide charge (net charge and charge density) stood out. In particular, descriptors associated with peptide charge were the most significant in both types of models. Net charge, charge density, charge at acidic pH, and isoelectric point suggest that a positive charge is a common requirement in most antimicrobial peptides. This finding is consistent with one of the main mechanisms of action of the peptides proposed, based on electrostatic interactions between the cationic charges of the peptide and the anionic charges of the bacterial membrane [42]. Furthermore, descriptors such as isotropic surface area and ISAECI index (related to steric effects and the ability to form local dipoles) indicate that antimicrobial peptides are characterized by their tendency to have bulky side chains with greater steric effects. This can be associated with the structural stability required for their activity. It was also identified that a low molecular weight is favorable, since it can allow peptides to cross barriers such as the cell wall and reach the bacterial membrane, facilitating greater interaction with it [43]. Hydrophobicity descriptors were also relevant, although to a lesser extent. These quantify the frequency and orientation of polar and non-polar residues and are essential for the insertion of peptides into the lipid bilayer, as well as for the formation of transmembrane pores that destabilize the electrochemical gradient and lead to bacterial death [44]. Interestingly, we observed that antimicrobial peptides tend to be less hydrophobic than those without antimicrobial activity. This could be explained by the need for antimicrobial peptides to also contain polar residues, which could facilitate specific interactions with components of the bacterial membrane or favor a more controlled permeabilization, without compromising selectivity or inducing non-specific aggregation. Descriptors assessing the propensity to form secondary structures (alpha helices and turns) were also well rated by the models. For example, levitt-alpha and QIAN880129 estimate the probability of alpha-helix formation, while CHOP780212 assesses the probability of absence of these structures. The models show that antimicrobial peptides tend to incorporate amino acids that favor these conformations, which is consistent with their mechanism of action [43]. In this sense, it is recommended to favor the incorporation of amino acids such as alanine, glutamic acid, leucine and methionine, and to avoid glycine, tyrosine, serine and proline, since the latter discourage the formation of these structures. In contrast, beta-sheet-related descriptors were considered in only five classification models and two regression models, suggesting a limited contribution to predicting antimicrobial activity. This is possibly because few AMPs exist with this type of conformation, although they could also form functional amphipathic structures. Finally, the frequency of occurrence of certain amino acids, such as serine (S) and lysine (K), the latter with a positive charge, was evaluated, being more relevant in the regression models [45]. Despite the lower relative importance in this study, it is highlighted that lysine content may play a key role in antimicrobial activity and should be considered in the design and evaluation of new peptides.

To assess model performance, we predicted the antimicrobial activity (logMIC) of peptides not included in the original QSAR training set. Using the general regression model trained on the full dataset, the correlation with experimental logMIC values was low (R^2^ = 0.459), likely due to the dataset’s heterogeneity, which includes diverse microorganisms. To improve generalization, models were then generated for specific bacterial groups. The Gram-negative model showed a higher correlation (R^2^ = 0.476), possibly due to greater structural homogeneity and a larger subset size. Conversely, the Gram-positive model showed reduced performance (R^2^ = 0.339), possibly due to insufficient peptide data and higher bacterial diversity. Further refinement by bacterial genera improved predictions: Escherichia (R^2^ = 0.547) and Bacillus (R^2^ = 0.574) models showed higher accuracy, suggesting that intra-genus homogeneity and data quantity enhance prediction. In contrast, Pseudomonas (R^2^ = 0.415) and Staphylococcus (R^2^ = 0.360) models did not show improvement, likely due to high diversity and limited data. Peptides are highly heterogeneous molecules, which limits the development of species-specific QSAR models due to data scarcity [40]. Although similarity clustering can enhance predictability, general classification models still showed strong performance (MCC = 0.662). The Random Forest-based model effectively predicts antimicrobial activity using features like physicochemical properties, hydrophobicity, and secondary structure propensity. Additionally, specific models targeting Gram-positive (MCC = 0.708), Gram-negative (MCC = 0.675), and the genera *Escherichia* (MCC = 0.754), *Staphylococcus* (MCC = 0.701), *Bacillus* (MCC = 0.746), and *Pseudomonas* (MCC = 0.755) achieved even better results. These improvements in Matthews correlation and accuracy (see Table 6) support the strategy of restricting datasets to enhance model performance.

In the final part of this work, approximately one million peptides were designed, and their antimicrobial activity was predicted using classification and regression models. From these predictions, the 10 peptides with the greatest antimicrobial potential were selected. The structures of these peptides were modeled, revealing the presence of total or partial alpha-helix secondary structures in most of them. This suggests that the presence of secondary structures, especially alpha helices—a feature linked to membrane disruption mechanisms—is a common trait among high-activity AMPs, aligning with key descriptors identified in our QSAR analysis (e.g., propensity for helical folding). To advance these in silico findings toward therapeutic applications, future work should focus on experimental validation and stability optimization, such as using stability-guided design strategies similar to those employed for oncolytic peptides like LTX-315 [46], or mirror-image phage display techniques to enhance proteolytic resistance [47]. Furthermore, targeted delivery systems [48] could be explored to improve the bioavailability and tissue specificity of these promising candidates. Future work includes generating experimental MIC data for *Enterobacteriaceae* to correlate these values with specific molecular descriptors and develop more accurate, group-specific QSAR models. This strategy may be extended to other bacterial groups to strengthen predictive capabilities. Although the tool was validated using independent test sets, further experimental validation is planned to enhance its applicability.

## 5. Conclusions

In this study, we constructed a database comprising over 35,000 records of antimicrobial activity corresponding to 4874 peptides, from which 56 predictive models were developed using machine learning techniques. Models based on Random Forest algorithms demonstrated the best overall performance, particularly in classification tasks (MCC = 0.662–0.755), outperforming the regression models. Further analysis revealed that higher net positive charge, lower molecular weight, reduced hydrophobicity, increased alpha-helical propensity, and a greater frequency of residues such as lysine and serine were associated with enhanced antimicrobial activity. Nearly one million in silico–designed peptides were evaluated using the top-performing models, identifying those with the greatest predicted potential, with MIC values ranging from 12.58 to 33.87 µg/mL. These promising peptides shared distinct structural and physicochemical characteristics that may serve as key determinants of activity, offering valuable insights for the rational design of novel antimicrobial peptides.

## Figures and Tables

**Figure 1 pharmaceutics-17-00993-f001:**
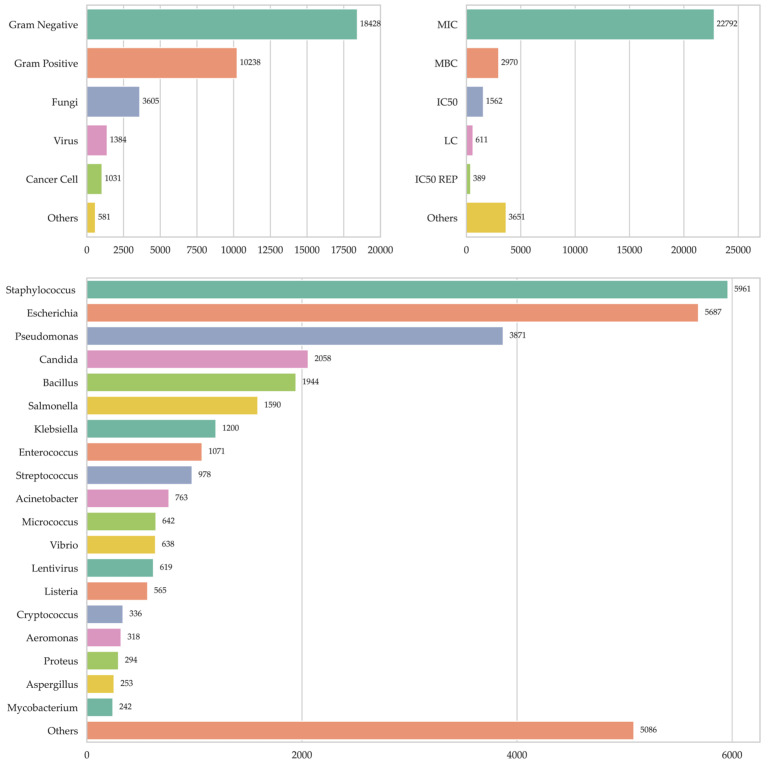
Number of records in the antimicrobial peptide database. Records by microorganism type at top left, records by antimicrobial activity measurement type at top right, and records by microorganism genus at bottom.

**Figure 2 pharmaceutics-17-00993-f002:**
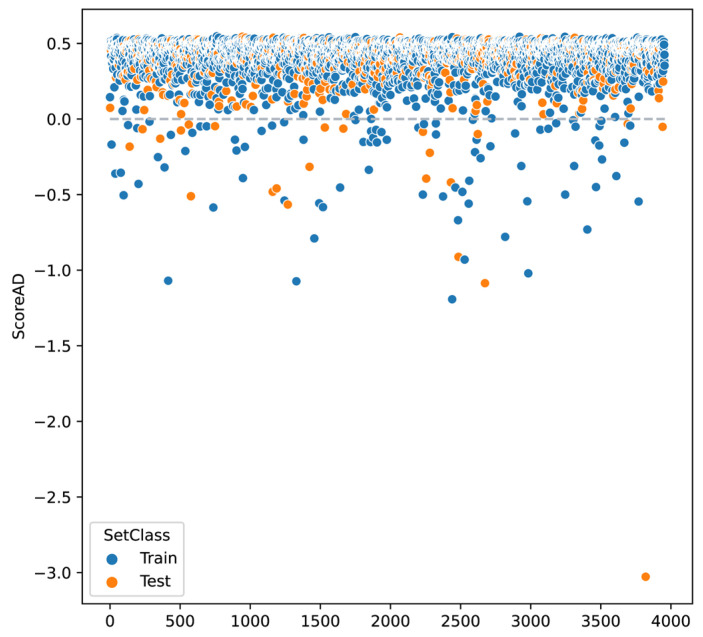
Distribution of the decision score for the applicability domain performed by k-NN on the general classification model. Values less than 0 are considered outliers and therefore outside the applicability domain. The orange indicates the observations corresponding to the test set, which defines the applicability domain, and the blue indicates the training set.

**Figure 3 pharmaceutics-17-00993-f003:**
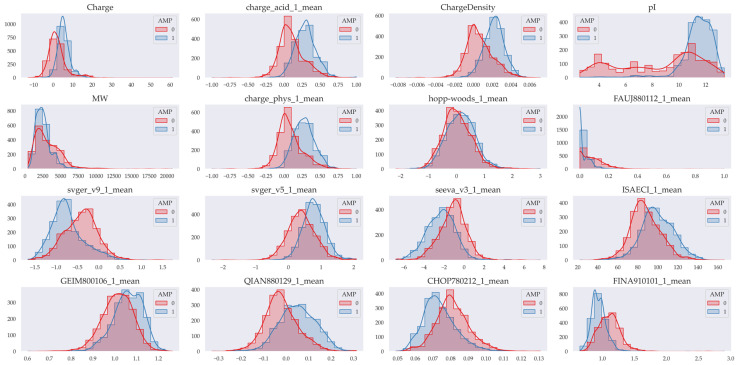
Distribution of some of the most important descriptors. AMPs (Class 1) are shown in blue, and non-AMPs (Class 0) are shown in red. The figure shows that the data have a differentiable distribution based on their class. The most relevant descriptors for the generated regression and classification models are shown.

**Figure 4 pharmaceutics-17-00993-f004:**
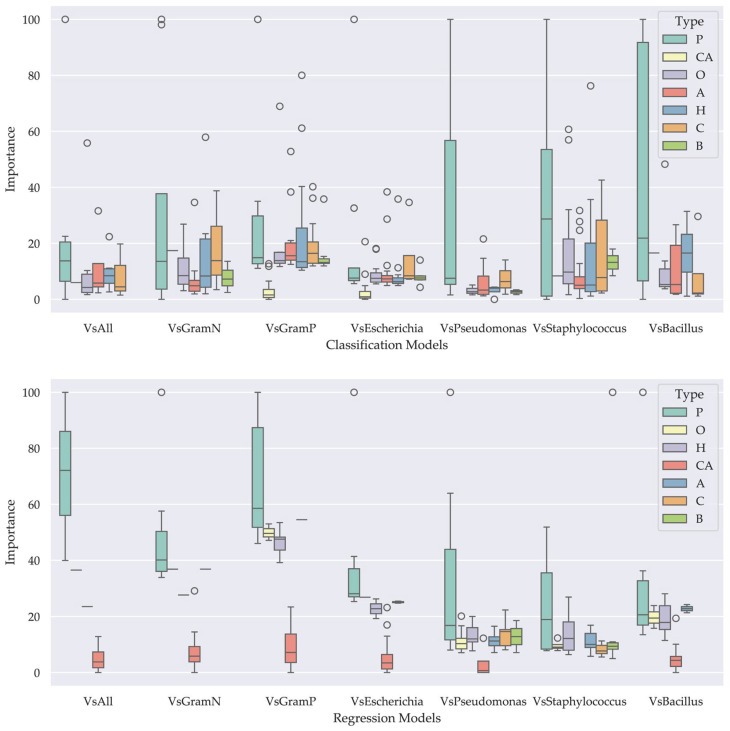
Box plots showing the importance of each descriptor type in the classification (**top**) and regression (**bottom**) models. The legend indicates the descriptor categories: Propensity for alpha-helical and coil structures (A), beta-sheet structures (B), general composition (C), amino acid composition (CA), hydrophobicity (H), other properties (O), and physicochemical properties (P).

**Figure 5 pharmaceutics-17-00993-f005:**
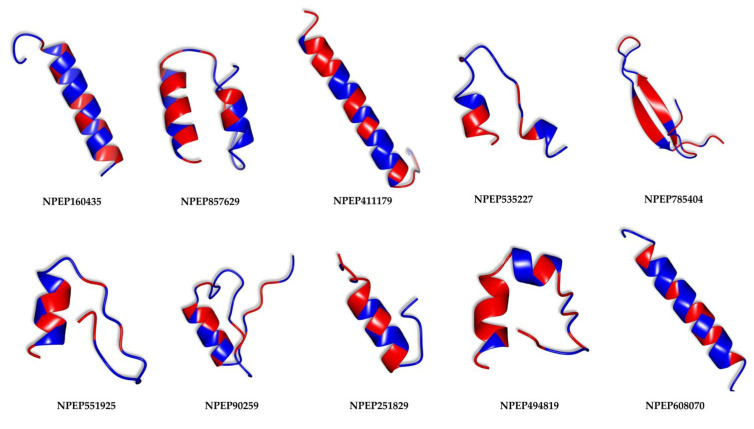
Secondary structures of peptides with the highest predicted antimicrobial potential. Hydrophobic residues are colored in red, while hydrophilic residues are shown in blue.

**Table 1 pharmaceutics-17-00993-t001:** Number of descriptors selected by the recursive elimination process (RFE) and genetic algorithm (GA) in each dataset.

	Regression			Classification		
Dataset	Initials	RFE	GA	Initials	RFE	GA
Peptides vs. All	221	66	39	250	60	26
Peptides vs. GramN	219	66	40	250	70	45
Peptides vs. GramP	219	52	41	250	245	118
Peptides vs. *Escherichia*	219	62	48	250	205	93
Peptides vs. *Pseudomonas*	219	64	64	250	55	28
Peptides vs. *Staphylococcus*	220	46	46	250	120	67
Peptides vs. *Bacillus*	218	40	40	250	55	32

**Table 2 pharmaceutics-17-00993-t002:** Final performance of the models on the test sets.

Dataset	${Regression}_{Test}$	${Classification}_{Test}$			
	** *R* ^2^ **	** *MCC* **	** *ACC* **	** *Pr* **	** *Sn* **
Peptides vs. All	0.459	0.662	0.831	0.838	0.829
Peptides vs. GramN	0.476	0.708	0.854	0.883	0.845
Peptides vs. GramP	0.339	0.675	0.837	0.825	0.863
Peptides vs. *Escherichia*	0.547	0.754	0.876	0.851	0.909
Peptides vs. *Pseudomonas*	0.415	0.755	0.877	0.856	0.911
Peptides vs. *Staphylococcus*	0.360	0.701	0.849	0.818	0.884
Peptides vs. *Bacillus*	0.574	0.746	0.873	0.902	0.860

**Table 3 pharmaceutics-17-00993-t003:** The average importance of variables in classification models based on *Random Forest*.

Classification Models	A	B	C	CA	H	O	P	*p*-Value **
vs. All	11.4	*	8.6	6.0	9.7	13.1	25.7	0.853
vs. GramN	7.3	7.8	17.9	17.4	17.1	11.3	31.3	0.349
vs. GramP	21.5	16.2	19.1	*2.5*	24.4	19.1	26.0	0.000
vs. *Escherichia*	10.3	8.1	14.6	*23*	9.1	9.1	18.6	0.000
vs. *Pseudomonas*	6.7	2.6	7.4	*	3.1	3.2	32.4	0.366
vs. *Staphylococcus*	8.9	13.2	15.8	8.4	15.3	18.6	34.0	0.755
vs. *Bacillus*	10.9	*	8.8	16.6	16.4	11.9	42.8	0.674

*: No variable of that type was considered by the model. ** α < 0.05, Kruskal–Wallis test, there is at least one significant difference between groups. A: Propensity for alpha-helical and twist structures; B: Propensity for beta-sheet structures; C: Composition; A: Amino acid composition; H: Hydrophobicity; O: Other properties; P: Physicochemical properties.

**Table 4 pharmaceutics-17-00993-t004:** The average of the importance of the variables in the regression models based on Random Forest.

Regression Models	A	B	C	CA	H	O	P	*p*-Value **
vs. All	*	*	*	3.79	23.53	36.59	72.12	0.005
vs. GramN	36.93	*	*	5.83	27.67	36.91	40.11	0.000
vs. GramP	54.52	*	*	7.16	47.56	49.60	58.55	0.000
vs. *Escherichia*	25.12	*	*	3.43	22.78	26.88	28.13	0.000
vs. *Pseudomonas*	11.25	12.84	14.61	0.68	11.99	10.31	16.81	0.042
vs. *Staphylococcus*	10.03	9.36	7.68	0.00	12.14	9.06	18.92	0.162
vs. *Bacillus*	22.80	*	*	4.36	17.89	19.46	20.63	0.000

*: No variable of that type was considered by the model. ** α < 0.05, Kruskal–Wallis test, there is at least one significant difference between groups. A: Propensity for alpha-helical and twist structures; B: Propensity for beta-sheet structures; C: Composition; A: Amino acid composition; H: Hydrophobicity; O: Other properties; P: Physicochemical properties.

**Table 5 pharmaceutics-17-00993-t005:** Sequence of peptides with the greatest antimicrobial potential.

Peptide	Sequence
NPEP160435	RNSIKPKVKKKWLKLTAKGLLLK
NPEP857629	KKLVKLKRGSQKKRILYYVGIPTLFKAAFKRISKL
NPEP411179	GIRIPKATSRKRYLKKIFEKKQAFVLFRFIP
NPEP535227	LFILNRAKRPKKSKIGLPRKITK
NPEP785404	TRVLGGTKKRRGVKLGAIKKWTLLGIRVLPQKR
NPEP551925	WRRLKLFKTSKVKYIYKKKGSLVK
NPEP90259	RLLNKPRKIKFVIIPRKPFKRKSFAIWTKLKLG
NPEP251829	AKVRLLITRLKKKFLIGRRDK
NPEP494819	AKIALLAPRITKKIKKFRFSAKKHFIF
NPEP608070	GRIRKTFLGLVGKKWYTKRIYSKR

**Table 6 pharmaceutics-17-00993-t006:** Properties and characteristics of the peptides with the greatest antimicrobial potential. AT: Alpha-helix trend, BT: Beta Trend-Turn, CT: Coil Trend, H: Hydrophobicity P: Polarity R: Refractivity: K: Number of lysines C: Net Charge.

Peptide	AT ^1^	BT ^1^	CT ^1^	H	P	R	K	C
NPEP160435	1.087	1.007	0.937	−0.160	8.520	18.553	8	9
NPEP857629	1.044	0.945	0.956	0.159	8.120	17.780	9	12
NPEP411179	1.131	0.912	0.913	−0.106	8.557	19.651	6	9
NPEP535227	1.019	1.082	1.004	−0.500	9.007	16.998	6	9
NPEP785404	1.018	0.998	0.961	0.018	8.258	16.364	6	10
NPEP551925	1.021	0.979	0.950	−0.234	8.488	19.639	8	10
NPEP90259	1.050	0.904	0.963	0.143	8.173	19.888	8	12
NPEP251829	1.117	0.881	0.902	−0.012	8.172	19.830	5	8
NPEP494819	1.121	0.902	0.927	−0.093	8.428	18.841	7	9
NPEP608070	0.991	0.969	0.971	0.146	8.060	19.420	5	9

^1^ Deleage and Roux scale.

## Data Availability

The antimicrobial peptide dataset is available as Appendix A. The complete data presented in this study are available upon request from the corresponding author. The machine learning models and prediction tool are available at: https://github.com/EliezerBonifacio/AMP_Prediction_ColabTool, accessed on 27 July 2025.

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
