# Peer review of "Predicting Antimicrobial Peptide Activity: A Machine Learning-Based Quantitative Structure–Activity Relationship Approach"

_pharmaceutics, 2025, doi:10.3390/pharmaceutics17080993_

Round 1

Reviewer 1 Report

Comments and Suggestions for Authors

The article by Eliezer Bonifacio and co-authors examines the prediction of antimicrobial peptide (AMP) activity using machine learning (ML) methods. The study highlights the relevance of the problem and demonstrates a deep understanding of ML methodologies.

The authors compared 56 ML models and determined the optimal one for this task. However, even models trained on specific groups of bacteria showed limited performance in regression tasks (R2 = 0.3390.574), which indicates an insufficient explanation for the variability of the data.

At the same time, this only confirms the complexity of the task, prediction of AMP properties in silico. I think the work remains valuable to researchers in this field.

As a small suggestion, I would recommend shortening the discussion section to eliminate redundancy and present ideas more concisely.

Author Response

Comments 1:  The article by Eliezer Bonifacio and co-authors examines the prediction of antimicrobial peptide (AMP) activity using machine learning (ML) methods. The study highlights the relevance of the problem and demonstrates a deep understanding of ML methodologies. The authors compared 56 ML models and determined the optimal one for this task. However, even models trained on specific groups of bacteria showed limited performance in regression tasks (R2 = 0.339–0.574), which indicates an insufficient explanation for the variability of the data. At the same time, this only confirms the complexity of the task, prediction of AMP properties in silico. I think the work remains valuable to researchers in this field. As a small suggestion, I would recommend shortening the discussion section to eliminate redundancy and present ideas more concisely.

Response 1: Thank you for pointing this out. We have reduced the content of the first, penultimate and antepenultimate paragraph, presenting the information of the regression models in a concise manner. Lines 375-388, 501-523.

Reviewer 2 Report

Comments and Suggestions for Authors

In this paper, a computational technique was applied to the analysis of potential new antimicrobial agents. The authors used ML to predict the activity of antimicrobial peptides. The study used logistic regression (LR), support vector machine (SVM), Random Forest (RF) and Gradient models, which were reasonable choice and effectively copes with the QSAR challenges.

1. While some peptides seem to be both active and inactive against different microorganisms, the rationale for averaging MIC results could be more robustly presented.

2. The main problem is that the authors calculate the logMIC. However, they should have investigated the actual MIC against selected bacteria, at least for several synthesized peptides. Without this, the current results are not scientifically validated in vitro. Thus, there is no confirmation whether the suggested AMP_Prediction_ColabTool actually works. Furthermore, MIC values ​​should be reported in ug/ml or mg/L. Unfortunately, reporting logMIC does not allow any comparison with real MIC studies in classical microbiology.

3. Single confirmatory MIC tests on bacteria and fungi should be performed and the methodology and results should be described.

4. Results and conclusions should be changed according to the MIC results obtained on microbiological cultures.

Author Response

In this paper, a computational technique was applied to the analysis of potential new antimicrobial agents. The authors used ML to predict the activity of antimicrobial peptides. The study used logistic regression (LR), support vector machine (SVM), Random Forest (RF) and Gradient models, which were reasonable choice and effectively copes with the QSAR challenges.

Comments 1:While some peptides seem to be both active and inactive against different microorganisms, the rationale for averaging MIC results could be more robustly presented.  

Response 1: We rewrited the section “2.3.3 Data preparation for classification models” to explain better the MIC averaging of some peptides. Lines 102-117.

Comments 2: The main problem is that the authors calculate the logMIC. However, they should have investigated the actual MIC against selected bacteria, at least for several synthesized peptides. Without this, the current results are not scientifically validated in vitro. Thus, there is no confirmation whether the suggested AMP_Prediction_ColabTool actually works. Furthermore, MIC values ​​should be reported in ug/ml or mg/L. Unfortunately, reporting logMIC does not allow any comparison with real MIC studies in classical microbiology. 

Response 2:  We added in supported material a table (table S6) with the actual experimental MIC data (in ug/ml) we have used for calculating logMIC for all combinations between peptides and bacterial. These data was used to create the models which are the basis for generating the AMP_Prediction_ColabTool. This tool is a product from the research described in this paper. We included in the supported material the specific references where we have taken the original MIC data.

Comments 3: Single confirmatory MIC tests on bacteria and fungi should be performed and the methodology and results should be described.

Response 3: The MIC data used for generation of QSAR models were obtained experimentally by several studies (supported material: table S6). However, we are planning in the future to generate our own experimental MIC data for Enterobacteriaceae group in order to relate their MIC values with specific descriptors which help us to generate better predictive QSAR specific models for this group.  Lines 537-542. 

Comments 4:  Results and conclusions should be changed according to the MIC results obtained on microbiological cultures.

Response 4: We added in the discussion section some comments about the possibility to continue this research generating experimentally data of some specific groups of bacterial. Lines 537-542.

Round 2

Reviewer 2 Report

Comments and Suggestions for Authors

The authors significantly corrected the manuscript according to the reviewer's suggestions. Recently, I recommend the article for publication.